# The Evolving Role of Immune-Checkpoint Inhibitors in Malignant Pleural Mesothelioma

**DOI:** 10.3390/jcm12051757

**Published:** 2023-02-22

**Authors:** Maxime Borgeaud, Floryane Kim, Alex Friedlaender, Filippo Lococo, Alfredo Addeo, Fabrizio Minervini

**Affiliations:** 1Oncology Department, University Hospital Geneva (HUG), 1205 Geneva, Switzerland; 2Oncology Department, Clinique Générale Beaulieu, 1206 Geneva, Switzerland; 3Department of Thoracic Surgery, Università Cattolica del Sacro Cuore, 00168 Rome, Italy; 4Thoracic Surgery, Fondazione Policlinico Universitario A. Gemelli Istituto di Ricovero e Cura a Carattere Scientifico (IRCCS), 00168 Rome, Italy; 5Division of Thoracic Surgery, Cantonal Hospital of Lucerne, 6000 Lucerne, Switzerland

**Keywords:** malignant pleural mesothelioma, immunotherapy, immune checkpoint inhibitors, biomarker mesothelioma

## Abstract

Malignant pleural mesothelioma (MPM) is a rare cancer usually caused by asbestos exposure and associated with a very poor prognosis. After more than a decade without new therapeutic options, immune checkpoint inhibitors (ICIs) demonstrated superiority over standard chemotherapy, with improved overall survival in the first and later-line settings. However, a significant proportion of patients still do not derive benefit from ICIs, highlighting the need for new treatment strategies and predictive biomarkers of response. Combinations with chemo-immunotherapy or ICIs and anti-VEGF are currently being evaluated in clinical trials and might change the standard of care in the near future. Alternatively, some non-ICI immunotherapeutic approaches, such as mesothelin targeted CAR-T cells or denditric-cells vaccines, have shown promising results in early phases of trials and are still in development. Finally, immunotherapy with ICIs is also being evaluated in the peri-operative setting, in the minority of patients presenting with resectable disease. The goal of this review is to discuss the current role of immunotherapy in the management of malignant pleural mesothelioma, as well as promising future therapeutic directions.

## 1. Introduction

Mesothelioma is a highly aggressive disease with poor prognosis. Malignant pleural mesothelioma (MPM) represents approximately 90% of all cases of mesothelioma. However, mesothelioma can also arise from mesothelium of other cavities, such as the peritoneum, in roughly 10% or, very rarely, the pericardium and tunica vaginalis [1]. The prognosis of MPM is poor, with 5 year all stages overall survival (OS) of approximately 10% [2]. The incidence of MPM has increased in recent decades [3]. This is partly explained by the lag-time between measures applied to decrease asbestos exposure and their impact. MPM often arises decades after exposure to asbestos [4]. Moreover, while the use of asbestosis has dramatically decreased since the 1990s in first-world countries [5], its use as an insulating material is still common in other parts of the world.

Immunotherapy with immune checkpoint inhibitors (ICI) has dramatically changed the landscape of management of MPM in the past few years (Table 1, Table 2 and Table 3). The aim of this article is to review this recent progress and provide future perspectives of immunotherapy in MPM.

## 2. Pathology, Molecular Biology and Tumor Micro-Environment

Histologically, MPM can be classified into three main subtypes: the epithelioid, sarcomatoid and biphasic (or mixed) subtypes. This histologic classification is of paramount importance as it has prognostic and therapeutic implications. Patients presenting with MPM of the sarcomatoid subtype have a worse prognosis when treated with chemotherapy, compared with epithelioid subtypes, with about half their median overall survival [6]. Biphasic MPM encompasses mesothelioma with both epithelioid and sarcomatoid characteristics. From a molecular perspective, MPMs are characterized by a relatively low frequency of mutations, and by copy number alterations. The most frequent alterations are found in BAP-1, CDKN2A/B, two tumor suppressor genes, and TP53 and NF-2 [7]. BAP-1 mutation is more frequent in epithelioid subtypes, whereas the loss of CDKN2A is more common in epithelioid subtypes, and is associated with a worse prognosis [8,9].

The tumor micro-environment in MPM is highly immunosuppressive. The majority of immune infiltrates in MPM is composed of tumor-associated macrophages with immunosuppressive proprieties, and myeloid-derived suppressors cells [10]. Cytotoxic T lymphocytes CD8+ represent a minority of the immune microenvironment [11]. Non-epithelioid MPM subtypes have been shown to have higher level of tumor-infiltrating cytotoxic T lymphocytes CD8+ [12]. Moreover, MPM tumors cells themselves exert an immune inhibitory effect through the expression of various immune checkpoints, such as PD-L1, TIGIT, VISTA and IDO1 among others [12,13]. PD-L1 positivity in tumor cells is more frequent in non-epithelioid MPM, where it is expressed in 30% of tumors, compared to 10–15% in epithelioid MPM. PD-L1 expression in tumor cells in MPM correlates with histological features of aggressiveness such as higher grade, high Ki67 index and necrosis, and thus worse response to chemotherapy [12].

## 3. Immune Checkpoints Inhibitors in the Management of Potentially Resectable Malignant Pleural Mesothelioma

The benefit of surgery in the management of MPM is controversial. There are no reliable randomized trial data to support this strategy over systemic treatment alone. Large retrospective series seem to show better outcome with surgery for the management of resectable MPM, when complete gross resection can be achieved [14,15]. However, retrospective data carry a major risk of selection bias (e.g., fitter patients undergoing surgery) and thus are of limited value in evaluating the benefit of a surgical approach. The MARS-1 trial was the first trial to randomize patients with resectable MPM between surgery and systemic treatment alone [16]. Due to poor accrual, the design of the study was changed to a feasibility study, assessing the achievability of randomizing patients between surgery or systemic treatment alone. The study did not show any difference in terms of OS but was underpowered to draw any conclusion. The MARS-2 trial, comparing extended pleural dissection with chemotherapy to chemotherapy alone, is currently ongoing to answer this question [17]. Full accrual has been reached, and the results are eagerly awaited. 

Of note, neither of these trials incorporated ICI as a systemic treatment component. A few phase I/II studies are testing the combination of ICI-chemotherapy in the peri-operative setting for MPM. The S1619 trial evaluated the association of atezolizumab with cisplatin-pemetrexed for four pre-operative cycles for potentially resectable MPM, followed by atezolizumab maintenance [18]. The combination was deemed safe and tolerable as no patient developed grade 4 or 5 immune-related adverse events. However, among the 28 patients included, 18 could undergo surgery. The phase III Atezo-Meso study is currently recruiting in Italy, and compares adjuvant atezolizumab with placebo among patients with resected MPM. Patient in this trial can receive chemotherapy as an adjuvant or neoadjuvant treatment [19].

Durvalumab and the combination of durvalumab and tremelimumab have recently been evaluated in the neoadjuvant setting in a small phase II study among 20 patients with resectable MPM [20]. The primary objective of the study was to determine the alteration of the intratumoral CD8/Treg ratio after neoadjuvant ICI. Both ICI regimens induced CD8 T cell infiltration into MPM tumors but did not alter CD8/Treg ratios. The combination of durvalumab and tremelimumab could induce a mobilization of CD57+ effector memory T cells from the bone marrow into the blood circulation and an increase in the formation of tertiary lymphoid structures in tumors that were rich in CD57+T cells. Interestingly, 85% of the patients underwent surgery, and 35% exhibited pathological evidence of tumor response, with 11.8% achieving major pathological response (>90% of tumor regression). Finally, patients receiving the double ICI combination had longer median overall survival than those receiving durvalumab alone (not reached versus 14 months). Of note, the combination was, unsurprisingly, associated with more adverse events than durvalumab monotherapy, with immune-related adverse event grade 3 or more occurring in 27% versus 8%, respectively. These results should be hypothesis generating, but are insufficient to draw further conclusions from such a small study. Perioperative approaches with nivolumab with or without ipilimumab are ongoing in phase I/II trials and may shed further light on this approach [21]. Interestingly, similar approaches are under investigation in resectable peritoneal mesothelioma, with a phase II trial of neoadjuvant ipilimumab and nivolumab [22].

## 4. Immune Checkpoints Inhibitors in the Management of Unresectable Malignant Pleural Mesothelioma

Most patients have an unresectable disease at initial presentation. For MPM, platinum and pemetrexed chemotherapy was the only approved first-line treatment regimen from 2004 [23], until recently. The addition of bevacizumab to the platinum-pemetrexed backbone was proven beneficial in the French phase III MAPS trial [24]. The addition of bevacizumab to pemetrexed plus cisplatin significantly improved OS with an increased rate of adverse events. Its implementation as a standard of care has varied worldwide, depending on approvals of bevacizumab in this setting by regulatory authorities. 

In the management of MPM, ICIs were first tested in the second- or third-line setting. The anti-CTLA-4 antibody tremelimumab was tested in the second-line setting in the phase IIb DETERMINE study, where it was compared to a placebo [25]. The trial was negative, with no difference in OS (7.7 vs. 7.3 months for tremelimumab and placebo, respectively). Pembrolizumab, an anti-PD-1, was tested as a monotherapy in the second or higher line setting in the phase Ib KEYNOTE-028 trial [26]. The overall response rate was 20% (five out of twenty-five patients), with a median duration of response of about a year. These results led to the phase III ETOP 9-15 PROMISE-meso trial, comparing pembrolizumab to chemotherapy of gemcitabine or vinorelbine, in the second-line setting, regardless of PD-L1 expression [27]. The trial was negative, with no difference in the primary outcome of PFS between both arms (2.5 vs. 3.4 months for pembrolizumab and chemotherapy, respectively, HR 1.06; 95% CI 0.73–1.53) or overall survival (HR 1.04; 95% CI 0.66–1.67). Interestingly, the overall response rate was higher with pembrolizumab than chemotherapy: 22 vs. 6%. However, the median duration of response was only 4.6 months with pembrolizumab. In the chemotherapy arm, 63% of patients crossed over to pembrolizumab, but there was no survival difference, even adjusting for cross-over. The ETOP 9-15 PROMISE-meso trial included a majority of patients with epithelioid subtypes, with only 11.1% of patients with non-epithelioid MPM. About 50% of patients had PD-L1 positive tumors. PD-L1 expression did not demonstrate any predictive value for pembrolizumab efficacy, nor any prognostic value in this trial.

Nivolumab, another anti-PD-1 antibody, was also tested for MPM in the relapse setting. The phase II MERIT trial evaluated nivolumab alone after progression on chemotherapy in Japanese patients. In this single arm trial, the overall response rate with nivolumab was 29%, with 17.3 months of median overall survival [28]. A European study showed similar results, in the same setting, with an ORR of 24% [29]. In both trials, PD-L1 expression in tumor cells did not correlate with the outcome. The phase III CONFIRM study was designed to confirm nivolumab efficacy in the second-line setting [30]. The CONFIRM trial randomized patients with pleural or peritoneal mesothelioma who had progressed after first-line platinum-based chemotherapy to nivolumab or placebo. There was an improvement in both co-primary endpoints, PFS and OS, in the nivolumab arm. The PFS was 3 vs. 1.8 months (HR 0.67, 95% CI 0.53–0.85; *p* = 0.0012) and OS 10 versus 6.9 months (HR 0.69, 95% CI 0.52–0.91; *p* = 0.009) in the nivolumab and placebo arms, respectively.

Following results observed in other tumor types with ICI combinations [31,32], the association of an anti-PD1 and an anti-CTLA-4 was evaluated in unresectable MPM. Durvalumab and tremelimumab were assessed in the single arm phase II study, NIBIT MESO1. In this study, 40 patients were included and treated in the first or second-line setting with durvalumab 20 mg/kg and tremelimumab 1 mg/kg, every four weeks for four cycles, followed by maintenance durvalumab for nine cycles. The ORR was 28%, with a median duration of response of 16.1 months and median OS of 16.6 months [33]. Moreover, 15% of patients were still alive at four years [34], suggesting that long disease control might be feasible in a fraction of patients with ICI combination, as seen in other tumor types. Larger confirmatory trials are needed.

The combination of nivolumab with ipilimumab was evaluated in the single arm phase II study, INITIATE [35]. Thirty-eight patients with MPM who had progressed after at least one line of platinum based chemotherapy received nivolumab 240 mg every two weeks and ipilimumab 1 mg/kg every six weeks. The ORR was 29%, with a median PFS of 6.2 months. Nivolumab and ipilimumab was also evaluated in the randomized phase II IFCT1501 MAPS2 trial [36]. This study enrolled 125 patients with MPM who had progressed after one or two lines of chemotherapy. Patients were randomized between nivolumab alone or the combination of nivolumab and ipilimumab. The trial was neither powered nor designed to compare these regimens head-to-head. The primary outcome of disease control rate was 50% in the nivolumab–ipilimumab arm and 44% in the nivolumab monotherapy arm. ORRs were 27.8% and 18·5% in the nivolumab–ipilimumab arm and nivolumab monotherapy arms, respectively. Median OS was 15.9 months in the combination arm and 11.9 months with nivolumab alone. The combination ICI was, unsurprisingly, associated with more toxicity, with 26% grade 3–4 treatment related adverse events versus 14% for nivolumab alone, and three toxic deaths.

Checkmate 743 was the paradigm changing trial in the management of unresectable MPM, establishing dual ICI as a standard of care [6]. Checkmate 743 was a Phase III randomized trial comparing nivolumab and ipilimumab to platinum-pemetrexed chemotherapy in the first-line treatment of unresectable MPM. Nivolumab was given at 3mg/kg IV every two weeks, while ipilimumab was given at 1mg/kg every six weeks. Immunotherapy was pursued until progression or unacceptable toxicity for up to two years. Chemotherapy, on the other hand, was given every three weeks (cisplatin 75 mg/m^2^ or carboplatin area under the curve 5, and pemetrexed 500 mg/m^2^) for up to six cycles. The study met its primary endpoint of median overall survival, with an improvement of four months, for the immunotherapy combination as compared with chemotherapy (18.1 months vs. 14.1 months; HR 0.74; 95% CI 0.6–0.91; *p* = 0.002). Of note, the relative benefit of ICI compared to chemotherapy seems to vary across different histologic subtypes, although this analysis was not preplanned. Among the 75% of patients with epithelioid subtypes in Checkmate 743, only a trend for OS benefit was observed without statistical significance (median OS: 18.7 months vs. 16.5 months; HR 0.69; 95% CI 0.69–1.08). On the other hand, for patients with non-epithelioid subtypes, ipilimumab and nivolumab led to a statistically significant prolonged median OS (18.1 months vs. 8.8 months; HR 0.69; 95% CI 0.31–0.68). For ICI, median OS was comparable across all histologic subtypes (median OS of 18.1 and 18.7 months), whereas with chemotherapy the non-epithelioid subtypes clearly had a worse prognosis compared to epithelioid subtypes (median OS of 8.8 vs. 16.5 months), reflecting chemoresistance for these histologies. Therefore, the histological subtype cannot be seen as a true predictive biomarker of ICI efficacy, as the greater benefit observed with ICI in the non-epithelioid subgroups is more a reflection of the chemoresistant nature of this subgroup. Another subgroup analysis in the Checkmate 743 study compared patients with and without PD-L1 expression. In patients with PD-L1 expression of 1% or higher in tumor cells nivolumab and ipilimumab led to a better median OS compared with chemotherapy (18.0 months vs. 13.3 months; HR 0.69; CI 95% 0.55–0.87), whereas in the PD-L1 negative subgroup no difference was noted between both treatment arms. As mentioned before, other studies in the later-line setting showed no predictive value of PD-L1 expression for ICI benefit in MPM. PD-L1 expression is a known negative prognostic factor in MPM, and the difference seen in the Checkmate study could reflect a worse prognosis of PD-L1 positive tumors treated with chemotherapy (e.g., non-epithelioid subtypes), rather than a predictive biomarker of ICI efficacy. Regarding the safety of nivolumab and ipilimumab, grade 3–4 adverse events occurred in 30.3% of patients with 23% of patients discontinuing ICI due to adverse events, compared with 32% of grade 3–4 adverse events and a discontinuation rate of 15% for chemotherapy. Based on these results, the FDA approved the combination of nivolumab plus ipilimumab as a first-line treatment for unresectable MPM in October 2020. Of note, the FDA approved a different dosage of nivolumab from the Checkmate 743, with a dose of nivolumab of 360 mg every three weeks along with ipilimumab 1 mg/kg every six weeks. The choice of ICI seems straightforward for patients with non-epithelioid subtypes, given the chemoresistant nature of the disease. For epithelioid histology, the benefit of nivolumab and ipilimumab over chemotherapy is less clear. No direct comparison exists between nivolumab-ipilimumab and the platinum-pemetrexed-bevacizumab combination, although an indirect comparison seems to show no clear difference [37]. Moreover, in Checkmate 743, disease progression within 12 weeks of treatment initiation was more common with ICI than with chemotherapy.

## 5. Immune Checkpoints Inhibitors in Combination with Chemotherapy

The next step for unresectable MPM could be chemo-immunotherapy, following what has been seen in non-small cell lung cancer. Some phase II data show promising preliminary results for the association of chemoimmunotherapy, with cisplatin-pemetrexed and durvalumab. The DREAM study evaluated the association of durvalumab 1125 mg every three weeks given concurrently with cisplatin and pemetrexed chemotherapy for up to six cycles, followed by durvalumab maintenance for up to one year [38]. The six-month PFS, the primary endpoint, was 57%. The same combination was evaluated in a second phase II single arm trial, PrE0505 [39]. Among 54 patients evaluated in the first line setting, the ORR was 56.4% (95% CI 42.3–69.7), the median PFS was 6.7 months (95% CI 6.1–8.4) and the median OS was 20.4 months. Following these promising data, the randomized phase III DREAM3R trial comparing durvalumab with cisplatin or carboplatin-pemetrexed chemotherapy with chemotherapy alone is ongoing [40].

Other chemoimmunotherapy trials are also being evaluated in phase III studies in the frontline setting. The international, multicenter IND227 trial evaluates the association of pembrolizumab-cisplatin-pemetrexed, compared to chemotherapy alone [41]. The ETOP BEAT meso study compares the addition of ICI with atezolizumab to the MAPS regimen of cisplatin or carboplatin-pemetrexed-bevacizumab [42]. The results are eagerly awaited, as they could be practice changing. The same regimen is being assessed in advanced peritoneal mesothelioma in a randomized phase II trial of neoadjuvant chemotherapy and bevacizumab, with or without atezolizumab [43]. Interestingly, a small phase II trial in unresectable peritoneal mesothelioma reported early but promising results with a combination of an anti-PD-L1 and anti-VEGF, with atezolizumab and bevacizumab, with an ORR of 40% among 20 patients [44].

## 6. Immune Checkpoints Inhibitors in Combination with Radiotherapy

Radiotherapy can be performed in mesothelioma patients as a part of multimodality treatment or in a palliative setting. With the increasing role of immunotherapy in the treatment of several tumors, including mesothelioma, it should be noted that radiation therapy can itself be an immunomodulator. Therefore, radiation therapy could potentially accomplish a synergistic effect along with check point inhibitors. At the same time, the use of ICI in combination with radiotherapy could lead to an enhanced pulmonary toxicity resulting in severe pneumonitis. However, clinical data on outcomes and side effects combining both therapies are, so far, lacking.

## 7. Challenges in Immunotherapy for Malignant Pleural Mesothelioma

As highlighted above, predictive biomarkers remain elusive in this rare disease. Given the lack of head-to-head comparisons between ICIs and chemotherapy with bevacizumab in the front-line setting, biomarkers would be of great use to individualize management. While PD-L1 expression is predictive of a certain degree of chemoresistance, there is room for improvement. Currently, non-epithelioid subtypes appear to derive less benefit from chemotherapy, making immunotherapy the preferred front-line approach. Beyond this, optimal management is uncertain and pros and cons of each treatment should be discussed with patients.

Next, ongoing neoadjuvant approaches may redefine operability in many patients. Should this be the case, there will be uncertainty about the ICI rechallenge upon progression. Management will likely be extrapolated from what is done in other settings, including NSCLC, where rechallenge depends on the interval between treatments [45].

Efforts are ongoing to identify MPM earlier in at-risk, asbestos-exposed populations. The goal would be downstaging and improving survival. One such potential approach is the use of circulating tumor cells and circulating tumor DNA. Different markers may provide hints for early diagnosis, including mesothelin, osteopontin, HMGB1 and fibulin-3 [46]. Should this prove effective, more focus could shift to neoadjuvant ICI approaches, discussed previously.

## 8. New Immunotherapeutic Approaches

One interesting way for improving the efficacy of immunotherapy in MPM could be the blockade of novel immune checkpoints along with the exploration of new combinations of immune checkpoint inhibitors. The presence of immunosuppressive cells in the tumor microenvironment in MPM, and the frequent expression of other immune inhibitory checkpoint points, further support this approach [12]. Some preclinical data have shown that the combination of PD-1 and LAG-3 blockade can enhance anti-tumor cytotoxic T-cells activity and reduce tumor growth in a MPM model [47]. However, as of today, there are no clinical data evaluating the combination of ICI other than anti-PD-1 and anti-CTLA-4 in MPM.

On the other hand, ICIs in combination with other partners are also being tested. As an example, anetumab ravtansine, an antibody targeting mesothelin, conjugated with the cytotoxic anti-tubulin drug ravtansine, is currently being evaluated in a phase I/II study in association with pembrolizumab [48]. The combination of pembrolizumab with lenvatinib, a multi-tyrosine kinase inhibitor, has also been explored in the phase II PEMMELA study, in the second-line setting, and the preliminary results were presented at the International Association for the Study of Lung Cancer 2022 World Conference [49]. This trial included 38 patients with recurrent MPM after previous chemotherapy, who received pembrolizumab given at a dose of 200 mg once every three weeks plus lenvatinib at 20 mg orally once a day, until disease progression, unacceptable toxicity or for up to two years. Interestingly, the majority (89.5) of the patients had an epithelioid MPM and 47.4% were PD-L1 positive. An interesting 39.5% of ORR was observed. Grade 3 or 4 treatment-related adverse events were observed in 26 patients, the most common being hypertension and anorexia. Overall 76% of patients required dose reduction or treatment discontinuation.

Other attempts in inducing an immune response in MPM have included vaccines as part of the strategy. One of these strategies used the re-inoculation of autologous dentritic cells exposed ex vivo to MPM apoptotic cells from a tumor lysate in order to elicit an immune response [50]. This dendritic cell-based therapy was able to elicit an immunological response to tumor cells in some mesothelioma patients. Early phases trials reported an interesting mOS duration of 27 months [51]. Whether these prolonged survivals are due to a true effect of the vaccine or just reflect a rigorous patient selection in these trials is unknown. A phase III trial, DENIM, is currently ongoing, comparing vaccination plus best supportive care to best supportive care alone after progression to first line chemotherapy [52].

Chimeric antigen receptor (CAR)-T cells therapy represents another promising approach to treat MPM. In a phase I study of mesothelin targeted CAR-T cells administered in the intrapleural space, along with intravenous pembrolizumab, in 18 patients with malignant pleural disease, the median overall survival was almost two years, with two patients having complete metabolic responses on PET-CT [53]. Fibroblast activating protein (FAP) represents another potential target for CAR T-cells in MPM. In early phase I results, FAP-targeted CAR-T appeared safe [54].

Other immunotherapeutic strategies based on local cytokine delivery through genetically modified viruses have been evaluated in MPM. Adenovirus-delivered interferon Alpha-2b (Ad-IFN) is a replication-defective adenoviral vector containing the human interferon-alpha2b gene. The intrapleural administration of Ad-IFN leads to the transfection of benign mesothelial and MPM cells, generating a large production of interferon within the pleural space and the tumor, resulting in a strong stimulus to the immune system [55]. The intrapleural administration of Ad-IFN with concomitant celecoxic and chemotherapy was evaluated in a cohort of 40 patients, with an ORR of 25% and a mOS of 21.5 months for patients receiving the combination in the second-line setting. Following these promising results, the phase III INFINITE trial was launched, assessing the efficacy of intrapleural Ad-IFN in combination with celecoxib and gemcitabine in the second- or third-line setting. The trial has terminated accrual and results are expected in 2024 [56].

Radiotherapy can be part of the treatment strategy of MPM, either as a palliative approach for pain management or in the peri operative setting [57]. In the peri-operative setting, radiation therapy has been used whether as an adjuvant treatment or as part of a neoadjuvant therapy, although little evidence supports this strategy [57]. Through the radiation-induced tumor cell death and the release of neo-antigen, radiation therapy can enhance the immune response both at the local site and outside the irradiated field, an effect known as the abscopal effect [58]. While observed in other tumor types, there is currently no evidence of a synergistic role of immune checkpoint inhibitors and radiation therapy in MPM, but several trials are ongoing [59,60,61].

## 9. Conclusions

Combination immunotherapy represents a new standard of care for patients with unresectable malignant pleural mesothelioma. Unfortunately, the proportion of MPM patients that achieve prolonged response and survival with immune checkpoint inhibitor remains low, and new strategies are awaited. The combination of chemotherapy with immune checkpoint inhibitors likely represents the next step. A better understanding of the mechanisms of immune resistance will be needed to adapt treatment strategies and improve outcomes.

## Figures and Tables

**Table 1 jcm-12-01757-t001:** Selected single arm trials of immune-checkpoint inhibitors in MPM.

Study	Phase	Intervention	Patients (n)	Setting	Results
KEYNOTE 028	Ib	Pembrolizumab 10 mg/kg q2 weekly	25	2nd or more	ORR: 20% mPFS 5.4 months mOS: 18 months
NCT02399371	II	Pembrolizumab 200 mg q3 weekly	65	2nd line	ORR: 19% mPFS: 4.5 months mOS: 11.5 months
MERIT	II	Nivolumab 240 mg q2 weekly	34	2nd line	ORR: 29% mPFS 6.1 months mOS: 17.3 months
INITIATE	II	Nivolumab 240 mg q2 weekly + ipilimumab 1 mg/kg q6 weekly	38	2nd line or more	ORR: 29% mPFS: 6.2 months mOS: NR.
NIBIT-MESO-1	II	Tremelimumab 1 mg/kg + durvalumab 20 mg/kg q4 weekly	40	1st or 2nd line	ORR: 28% mPFS 5.7months mOS 16.6 months

**Table 2 jcm-12-01757-t002:** Completed randomized trials of immunotherapy in MPM.

Study	Phase	Intervention	Patients (n)	Setting	Results
CHECKMATE 743	III	Nivolumab + ipilimumab vs. chemotherapy	605	Unresectable disease 1st line	OS: 18.1 vs. 14.1 months mPFS: 6.8 vs. 7.2 months ORR: 39.6% vs. 44.0%
MAPS2	II	Nivolumab, or nivolumab + ipilimumab (non-comparative study)	125	Unresectable disease, 2nd line	OS: 11.9 months (nivolumab), 15.9 months (combination) ORR: 16.7% (nivolumab) and 25.9% (combination) mPFS: 4 months (nivolumab) and 5.6 months (combination)
CONFIRM	III	Nivolumab vs. placebo	332	Unresectable disease, 2nd line	OS: 10.2 vs. 6.9 months mPFS: 3 vs. 1.8 months
PROMISE-meso ETOP 9-15	III	Pembrolizumab vs. chemotherapy	144	Unresectable disease, 2nd line	OS 10.7 vs. 12.4 months (negative) mPFS: 2.5 vs. 3.4 months ORR: 22% vs. 6%
DETERMINE	III	Tremelimumab vs. placebo	571	Unresectable disease, 2nd line	OS: 7.2 vs. 7.7 months (negative) mPFS: 7.7 vs. 7.3 months
NCT02592551	II	Durvalumab + tremelimumab vs. durvalumab, followed by surgery	20	Resectable disease, neoadjuvant	Major pathological response: 11.8% (combination) OS: Not reached vs. 14 months

**Table 3 jcm-12-01757-t003:** Early phases results of promising approaches in MPM.

Study	Phase	Intervention	Patients (n)	Setting	Results
DREAM	II	Durvalumab + platinum-Pemetrexed	54	Unresectable disease, 1st line	OS: 18.4 months
PrE0505	II	Durvalumab + cisplatin-pemetrexed	55	Unresectable disease, 1st line	ORR: 56.4% OS duration: 20.4 months
NCT04577326	I	Mesothelin-targeted CAR -T cells and pembrolizumab	25	Unresectable disease, 2nd line	OS: 23.9 months
PMR-MM-002 (NCT01241682)	I	Dendritic-cell-based vaccine and low-dose cyclophosphamide	10	Unresectable disease, 2nd line	OS: 70% ≥ 24 months

## Data Availability

Not applicable.

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
