# Peer review of "The Evolving Role of Immune-Checkpoint Inhibitors in Malignant Pleural Mesothelioma"

_jcm, 2023, doi:10.3390/jcm12051757_

Round 1
Reviewer 1 Report
This is a very good review of challenging pathology that has limited published data to guide the treatment team. This broad overview is clinically relevant and well written. I think additional comments, even if brief, regarding the role of radiation therapy could also make this more relevant. The data for radiation are also poor, but radiation is often given in the perioperative and definitive settings. Additionally there is some limited data that radiation may serve as an immunomuldator with improved response rates in the setting of lung cancer. It may be interesting to briefly comment on this as a future area of exploration.
Author Response
Dear Reviewer,
thanks a lot for your suggestion. We added a paragraph focusing the attention of the combination therapy (ICI + radiotherapy).
Reviewer 2 Report
The authors have provided a comprehensive review of ICI treatment in mesothelioma.
-Line 63-64: 'As discussed later, PD-L1 expression on tumor cells in MPM is a negative prognostic biomarker'. Would be careful about this association since PDL1 expression is more common in sarcomatoid histology, which is also associated with a worse prognosis.
-Line 85: 'However, among the 28 patients included, only 18 could undergo surgery.' This is a similar rate to neoadjuvant chemotherapy trials so would not make much of it.
-Line 207: ', and the difference seen in the Checkmate study could reflect a worse prognosis of PD-L1 positive tumors treated with chemotherapy, rather than a predictive biomarker of ICI efficacy.' Or could be that more PDL1 positive tumors were non-epithelioid, which did better with ICI compared to chemo, as you point out.
-Could point out that the dose of ipilimumab/nivolumab approved by the FDA is different from what is tested (flat dosing of nivolumab every 3 weeks with ipilimumab every 6 weeks).
-Line 258: 'Management will likely be extrapolated from what is done in other settings, includ ing NSCLC, where rechallenge depends on the interval between treatments'. Not sure we have a good rubric for ICI re-challenge. Please provide reference if you keep this statement.
- line 279: 'The combination of pembrolizumab with lenvatinib, a multi-tyrosine kinase inhibitor is also being explored in a phase II study, in the second line setting' There is data from this that was presented at IASLC world lung in 2022. ORR 39.5%
- Would consider a supplemental table of other trials mentioned in the paper (neoadjuvant trials?).
Author Response
Dear Reviewer,
thanks a lot for your time to review our manuscript.
We revised the manuscript following all your suggestions.
thanks!
Reviewer 3 Report
Dr. Maxime Borgeaud et al. reported a paper summarizing the treatment of MPM, especially ICI. This review article is an excellent summary of the clinical trials and other issues I would like to refer to in the future. I will list a few minor revisions; however, this paper is worthy.
My comments are listed below.
Minor comments:
1. Why did the authors decide to include only randomized controlled trials in Table A? I believe there are essential single-arm trials, such as the MERIT trial, so there is no need to list only randomized controlled trials in the Table. The authors need to reconsider the subject to be listed in the Table.
2. For the clinical trials listed in Table A, the authors should include ORR, PFS (or EFS, DFS), and OS, if the results are all known.
3. Please correct the "XXmg/m2" used in the text to "XX mg/m2".
Author Response

(The authors gave the same response as above.)
